# Dendritic cortical microcircuits
# approximate the backpropagation algorithm

**João Sacramento**[*]
Department of Physiology
University of Bern, Switzerland
sacramento@pyl.unibe.ch

**Rui Ponte Costa**[†]
Department of Physiology
University of Bern, Switzerland
costa@pyl.unibe.ch

**Yoshua Bengio**[‡]
Mila and Université de Montréal, Canada
yoshua.bengio@mila.quebec

**Walter Senn**
Department of Physiology
University of Bern, Switzerland
senn@pyl.unibe.ch

## Abstract

Deep learning has seen remarkable developments over the last years, many of them inspired by neuroscience. However, the main learning mechanism behind these advances – error backpropagation – appears to be at odds with neurobiology. Here, we introduce a multilayer neuronal network model with simplified dendritic compartments in which error-driven synaptic plasticity adapts the network towards a global desired output. In contrast to previous work our model does not require separate phases and synaptic learning is driven by local dendritic prediction errors continuously in time. Such errors originate at apical dendrites and occur due to a mismatch between predictive input from lateral interneurons and activity from actual top-down feedback. Through the use of simple dendritic compartments and different cell-types our model can represent both error and normal activity within a pyramidal neuron. We demonstrate the learning capabilities of the model in regression and classification tasks, and show analytically that it approximates the error backpropagation algorithm. Moreover, our framework is consistent with recent observations of learning between brain areas and the architecture of cortical microcircuits. Overall, we introduce a novel view of learning on dendritic cortical circuits and on how the brain may solve the long-standing synaptic credit assignment problem.

## 1 Introduction

Machine learning is going through remarkable developments powered by deep neural networks (Le-Cun et al., 2015). Interestingly, the workhorse of deep learning is still the classical backpropagation of errors algorithm (backprop; Rumelhart et al., 1986), which has been long dismissed in neuroscience on the grounds of biological implausibility (Grossberg, 1987; Crick, 1989). Irrespective of such concerns, growing evidence demonstrates that deep neural networks outperform alternative frameworks in accurately reproducing activity patterns observed in the cortex (Lillicrap and Scott, 2013; Yamins et al., 2014; Khaligh-Razavi and Kriegeskorte, 2014; Yamins and DiCarlo, 2016; Kell et al., 2018). Although recent developments have started to bridge the gap between neuroscience

---

[*]Present address: Institute of Neuroinformatics, University of Zürich and ETH Zürich, Zürich, Switzerland
[†]Present address: Computational Neuroscience Unit, Department of Computer Science, SCEEM, Faculty of Engineering, University of Bristol, United Kingdom
[‡]CIFAR Senior Fellow

and artificial intelligence (Marblestone et al., 2016; Lillicrap et al., 2016; Scellier and Bengio, 2017; Costa et al., 2017; Guerguiev et al., 2017), how the brain could implement a backprop-like algorithm remains an open question.

In neuroscience, understanding how the brain learns to associate different areas (e.g., visual and motor cortices) to successfully drive behaviour is of fundamental importance (Petreanu et al., 2012; Manita et al., 2015; Makino and Komiyama, 2015; Poort et al., 2015; Fu et al., 2015; Pakan et al., 2016; Zmarz and Keller, 2016; Attinger et al., 2017). However, how to correctly modify synapses to achieve this has puzzled neuroscientists for decades. This is often referred to as the synaptic credit assignment problem (Rumelhart et al., 1986; Sutton and Barto, 1998; Roelfsema and van Ooyen, 2005; Friedrich et al., 2011; Bengio, 2014; Lee et al., 2015; Roelfsema and Holtmaat, 2018), for which the backprop algorithm provides an elegant solution.

Here we propose that the prediction errors that drive learning in backprop are encoded at distal dendrites of pyramidal neurons, which receive top-down input from downstream brain areas (we interpret a brain area as being equivalent to a layer in machine learning) (Petreanu et al., 2009; Larkum, 2013). In our model, these errors arise from the inability to exactly match via lateral input from local interneurons (e.g. somatostatin-expressing; SST) the top-down feedback from downstream cortical areas. Learning of bottom-up connections (i.e., feedforward weights) is driven by such error signals through local synaptic plasticity. Therefore, in contrast to previous approaches (Marblestone et al., 2016), in our framework a given neuron is used simultaneously for activity propagation (at the somatic level), error encoding (at distal dendrites) and error propagation to the soma without the need for separate phases.

We first illustrate the different components of the model. Then, we show analytically that under certain conditions learning in our network approximates backpropagation. Finally, we empirically evaluate the performance of the model on nonlinear regression and recognition tasks.

## 2 Error-encoding dendritic cortical microcircuits

### 2.1 Neuron and network model

Building upon previous work (Urbanczik and Senn, 2014), we adopt a simplified multicompartment neuron and describe pyramidal neurons as three-compartment units (schematically depicted in Fig. 1A). These compartments represent the somatic, basal and apical integration zones that characteristically define neocortical pyramidal cells (Spruston, 2008; Larkum, 2013). The dendritic structure of the model is exploited by having bottom-up and top-down synapses converging onto separate dendritic compartments (basal and distal dendrites, respectively), a first approximation in line with experimental observations (Spruston, 2008) and reflecting the preferred connectivity patterns of cortico-cortical projections (Larkum, 2013).

Consistent with the connectivity of SST interneurons (Urban-Ciecko and Barth, 2016), we also introduce a second population of cells within each hidden layer with both lateral and cross-layer connectivity, whose role is to *cancel the top-down input* so as to leave only the backpropagated errors as apical dendrite activity. Modelled as two-compartment units (depicted in red, Fig. 1A), such interneurons are predominantly driven by pyramidal cells within the same layer through weights $\mathbf{W}_{k,k}^{\mathrm{IP}}$, and they project back to the apical dendrites of the same-layer pyramidal cells through weights $\mathbf{W}_{k,k}^{\mathrm{PI}}$ (Fig. 1A). Additionally, cross-layer feedback onto SST cells originating at the next upper layer $k+1$ provide a weak nudging signal for these interneurons, modelled after Urbanczik and Senn (2014) as a conductance-based somatic input current. We modelled this weak top-down nudging on a one-to-one basis: each interneuron is nudged towards the potential of a corresponding upper-layer pyramidal cell. Although the one-to-one connectivity imposes a restriction in the model architecture, this is to a certain degree in accordance with recent monosynaptic input mapping experiments show that SST cells in fact receive top-down projections (Leinweber et al., 2017), that according to our proposal may encode the weak interneuron 'teaching' signals from higher to lower brain areas.

The somatic membrane potentials of pyramidal neurons and interneurons evolve in time according to

$$\frac{d}{dt}\mathbf{u}_k^{\mathrm{P}}(t) = -g_{\mathrm{lk}}\,\mathbf{u}_k^{\mathrm{P}}(t) + g_{\mathrm{B}}\left(\mathbf{v}_{\mathrm{B},k}^{\mathrm{P}}(t) - \mathbf{u}_k^{\mathrm{P}}(t)\right) + g_{\mathrm{A}}\left(\mathbf{v}_{\mathrm{A},k}^{\mathrm{P}}(t) - \mathbf{u}_k^{\mathrm{P}}(t)\right) + \sigma\,\boldsymbol{\xi}(t) \qquad (1)$$

$$\frac{d}{dt}\mathbf{u}_k^{\mathrm{I}}(t) = -g_{\mathrm{lk}}\,\mathbf{u}_k^{\mathrm{I}}(t) + g_{\mathrm{D}}\left(\mathbf{v}_k^{\mathrm{I}}(t) - \mathbf{u}_k^{\mathrm{I}}(t)\right) + \mathbf{i}_k^{\mathrm{I}}(t) + \sigma\,\boldsymbol{\xi}(t), \qquad (2)$$

with one such pair of dynamical equations for every hidden layer $0 < k < N$; input layer neurons are indexed by $k = 0$, $g$'s are fixed conductances, $\sigma$ controls the amount of injected noise. Basal and apical dendritic compartments of pyramidal cells are coupled to the soma with effective transfer conductances $g_{\mathrm{B}}$ and $g_{\mathrm{A}}$, respectively. Subscript lk is for leak, A is for apical, B for basal, D for dendritic, superscript I for inhibitory and P for pyramidal neuron. Eqs. 1 and 2 describe standard conductance-based voltage integration dynamics, having set membrane capacitance to unity and resting potential to zero for clarity. Background activity is modelled as a Gaussian white noise input, $\boldsymbol{\xi}$ in the equations above. To keep the exposition brief we use matrix notation, and denote by $\mathbf{u}_k^{\mathrm{P}}$ and $\mathbf{u}_k^{\mathrm{I}}$ the vectors of pyramidal and interneuron somatic voltages, respectively. Both matrices and vectors, assumed column vectors by default, are typed in boldface here and throughout. Dendritic compartmental potentials are denoted by $\mathbf{v}$ and are given in instantaneous form by

$$\mathbf{v}_{\mathrm{B},k}^{\mathrm{P}}(t) = \mathbf{W}_{k,k-1}^{\mathrm{PP}} \, \phi(\mathbf{u}_{k-1}^{\mathrm{P}}(t)) \tag{3}$$

$$\mathbf{v}_{\mathrm{A},k}^{\mathrm{P}}(t) = \mathbf{W}_{k,k+1}^{\mathrm{PP}} \, \phi(\mathbf{u}_{k+1}^{\mathrm{P}}(t)) + \mathbf{W}_{k,k}^{\mathrm{PI}} \, \phi(\mathbf{u}_k^{\mathrm{I}}(t)), \tag{4}$$

where $\phi(\mathbf{u})$ is the neuronal transfer function, which acts componentwise on $\mathbf{u}$.

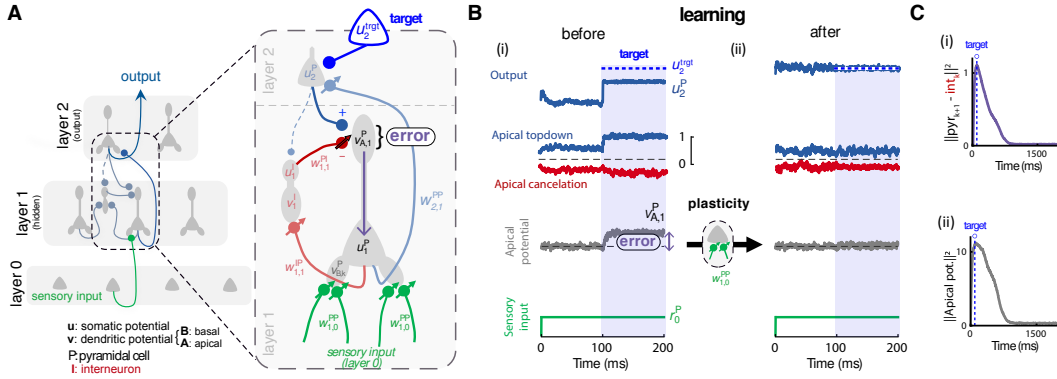

Figure 1: **Learning in error-encoding dendritic microcircuit network.** (**A**) Schematic of network with pyramidal cells and lateral inhibitory interneurons. Starting from a self-predicting state – see main text and supplementary material (SM) – when a novel teaching (or associative) signal is presented at the output layer ($\mathbf{u}_2^{\mathrm{trgt}}$), a prediction error in the apical compartments of pyramidal neurons in the upstream layer (layer 1, 'error') is generated. This error appears as an apical voltage deflection that propagates down to the soma (purple arrow) where it modulates the somatic firing rate, which in turn leads to plasticity at bottom-up synapses (bottom, green). (**B**) Activity traces in the microcircuit before and after a new teaching signal is learned. (i) Before learning: a new teaching signal is presented ($\mathbf{u}_2^{\mathrm{trgt}}$), which triggers a mismatch between the top-down feedback (grey blue) and the cancellation given by the lateral interneurons (red). (ii) After learning (with plasticity at the bottom-up synapses ($\mathbf{W}_{1,0}^{\mathrm{PP}}$)), the network successfully predicts the new teaching signal, reflected on no distal 'error' (top-down and lateral interneuron input cancel each other). (**C**) Interneurons learn to predict the backpropagated activity (i), while simultaneously silencing the apical compartment (ii), even though the pyramidal neurons remain active (not shown).

For simplicity, we reduce pyramidal output neurons to two-compartment cells: the apical compartment is absent ($g_{\mathrm{A}} = 0$ in Eq. 1) and basal voltages are as defined in Eq. 3. Although the design can be extended to more complex morphologies, in the framework of dendritic predictive plasticity two compartments suffice to compare desired target with actual prediction. Synapses proximal to the soma of output neurons provide direct external teaching input, incorporated as an additional source of current $\mathbf{i}_N^{\mathrm{P}}$. In practice, one can simply set $\mathbf{i}_N^{\mathrm{P}} = g_{\mathrm{som}}(\mathbf{u}_N^{\mathrm{trgt}} - \mathbf{u}_N^{\mathrm{P}})$, with some fixed somatic nudging conductance $g_{\mathrm{som}}$. This can be modelled closer to biology by explicitly setting the somatic excitatory and inhibitory conductance-based inputs (Urbanczik and Senn, 2014). For a given output neuron, $i_N^{\mathrm{P}}(t) = g_{\mathrm{exc},N}^{\mathrm{P}}(t) \left(E_{\mathrm{exc}} - u_N^{\mathrm{P}}(t)\right) + g_{\mathrm{inh},N}^{\mathrm{P}}(t) \left(E_{\mathrm{inh}} - u_N^{\mathrm{P}}(t)\right)$, where $E_{\mathrm{exc}}$ and $E_{\mathrm{inh}}$ are excitatory and inhibitory synaptic reversal potentials, respectively, where the inputs are balanced according to

$g^{\mathrm{P}}_{\mathrm{exc},N} = g_{\mathrm{som}} \frac{u^{\mathrm{trgt}}_N - E_{\mathrm{inh}}}{E_{\mathrm{exc}} - E_{\mathrm{inh}}}$, $g^{\mathrm{P}}_{\mathrm{inh},N} = -g_{\mathrm{som}} \frac{u^{\mathrm{trgt}}_N - E_{\mathrm{exc}}}{E_{\mathrm{exc}} - E_{\mathrm{inh}}}$. The point at which no current flows, $i^{\mathrm{P}}_N = 0$, defines the target teaching voltage $u^{\mathrm{trgt}}_N$ towards which the neuron is nudged[4].

Interneurons are similarly modelled as two-compartment cells, cf. Eq. 2. Lateral dendritic projections from neighboring pyramidal neurons provide the main source of input as

$$\mathbf{v}^{\mathrm{I}}_k(t) = \mathbf{W}^{\mathrm{IP}}_{k,k} \, \phi(\mathbf{u}^{\mathrm{P}}_k(t)), \tag{5}$$

whereas cross-layer, top-down synapses define the teaching current $\mathbf{i}^{\mathrm{I}}_k$. This means that an interneuron at layer $k$ permanently (i.e., when learning or performing a task) receives balanced somatic teaching excitatory and inhibitory input from a pyramidal neuron at layer $k+1$ on a one-to-one basis (as above, but with $\mathbf{u}^{\mathrm{P}}_{k+1}$ as target). With this setting, the interneuron is nudged to follow the corresponding next layer pyramidal neuron. See SM for detailed parameters.

## 2.2 Synaptic learning rules

The synaptic learning rules we use belong to the class of dendritic predictive plasticity rules (Urbanczik and Senn, 2014; Spicher et al., 2018) that can be expressed in its general form as

$$\frac{d}{dt} w = \eta \, (\phi(u) - \phi(v)) \, r, \tag{6}$$

where $w$ is an individual synaptic weight, $\eta$ is a learning rate, $u$ and $v$ denote distinct compartmental potentials, $\phi$ is a rate function, and $r$ is the presynaptic input. Eq. 6 was originally derived in the light of reducing the prediction error of somatic spiking, when $u$ represents the somatic potential and $v$ is a function of the postsynaptic dendritic potential.

In our model the plasticity rules for the various connection types are:

$$\frac{d}{dt} \mathbf{W}^{\mathrm{PP}}_{k,k-1} = \eta^{\mathrm{PP}}_{k,k-1} \left( \phi(\mathbf{u}^{\mathrm{P}}_k) - \phi(\hat{\mathbf{v}}^{\mathrm{P}}_{\mathrm{B},k}) \right) \left( \mathbf{r}^{\mathrm{P}}_{k-1} \right)^T, \tag{7}$$

$$\frac{d}{dt} \mathbf{W}^{\mathrm{IP}}_{k,k} = \eta^{\mathrm{IP}}_{k,k} \left( \phi(\mathbf{u}^{\mathrm{I}}_k) - \phi(\hat{\mathbf{v}}^{\mathrm{I}}_k) \right) \left( \mathbf{r}^{\mathrm{P}}_k \right)^T, \tag{8}$$

$$\frac{d}{dt} \mathbf{W}^{\mathrm{PI}}_{k,k} = \eta^{\mathrm{PI}}_{k,k} \left( \mathbf{v}_{\mathrm{rest}} - \mathbf{v}^{\mathrm{P}}_{\mathrm{A},k} \right) \left( \mathbf{r}^{\mathrm{I}}_k \right)^T, \tag{9}$$

where $(\cdot)^T$ denotes vector transpose and $\mathbf{r}_k \equiv \phi(\mathbf{u}_k)$ the layer $k$ firing rates. The synaptic weights evolve according to the product of dendritic prediction error and presynaptic rate, and can undergo both potentiation or depression depending on the sign of the first factor (i.e., the prediction error).

For basal synapses, such prediction error factor amounts to a difference between postsynaptic rate and a local dendritic estimate which depends on the branch potential. In Eqs. 7 and 8, $\hat{\mathbf{v}}^{\mathrm{P}}_{\mathrm{B},k} = \frac{g_{\mathrm{B}}}{g_{\mathrm{lk}} + g_{\mathrm{B}} + g_{\mathrm{A}}} \mathbf{v}^{\mathrm{P}}_{\mathrm{B},k}$ and $\hat{\mathbf{v}}^{\mathrm{I}}_k = \frac{g_{\mathrm{D}}}{g_{\mathrm{lk}} + g_{\mathrm{D}}} \mathbf{v}^{\mathrm{I}}_k$ take into account dendritic attenuation factors of the different compartments. On the other hand, the plasticity rule (9) of lateral interneuron-to-pyramidal synapses aims to silence (i.e., set to resting potential $\mathbf{v}_{\mathrm{rest}} = \mathbf{0}$, here and throughout zero for simplicity) the apical compartment; this introduces an attractive state for learning where the contribution from interneurons balances (or cancels out) top-down dendritic input. This learning rule of apical-targeting interneuron synapses can be thought of as a dendritic variant of the homeostatic inhibitory plasticity proposed by Vogels et al. (2011); Luz and Shamir (2012).

In experiments where the top-down connections are plastic, the weights evolve according to

$$\frac{d}{dt} \mathbf{W}^{\mathrm{PP}}_{k,k+1} = \eta^{\mathrm{PP}}_{k,k+1} \left( \phi(\mathbf{u}^{\mathrm{P}}_k) - \phi(\hat{\mathbf{v}}^{\mathrm{P}}_{\mathrm{TD},k}) \right) \left( \mathbf{r}^{\mathrm{P}}_{k+1} \right)^T, \tag{10}$$

with $\hat{\mathbf{v}}^{\mathrm{P}}_{\mathrm{TD},k} = \mathbf{W}_{k,k+1} \mathbf{r}^{\mathrm{P}}_{k+1}$. An implementation of this rule requires a subdivision of the apical compartment into a distal part receiving the top-down input (with voltage $\hat{\mathbf{v}}^{\mathrm{P}}_{\mathrm{TD},k}$) and another distal compartment receiving the lateral input from the interneurons (with voltage $\mathbf{v}^{\mathrm{P}}_{\mathrm{A},k}$).

## 2.3 Comparison to previous work

It has been suggested that error backpropagation could be approximated by an algorithm that requires alternating between two learning phases, known as contrastive Hebbian learning (Ackley et al., 1985). This link between the two algorithms was first established for an unsupervised learning task (Hinton and McClelland, 1988) and later analyzed (Xie and Seung, 2003) and generalized to broader classes of models (O'Reilly, 1996; Scellier and Bengio, 2017).

The concept of apical dendrites as distinct integration zones, and the suggestion that this could simplify the implementation of backprop has been previously made (Körding and König, 2000, 2001). Our microcircuit design builds upon this view, offering a concrete mechanism that enables apical error encoding. In a similar spirit, two-phase learning recently reappeared in a study that exploits dendrites for deep learning with biological neurons (Guerguiev et al., 2017). In this more recent work, the temporal difference between the activity of the apical dendrite in the presence and in the absence of the teaching input represents the error that induces plasticity at the forward synapses. This difference is used directly for learning the bottom-up synapses without influencing the somatic activity of the pyramidal cell. In contrast, we postulate that the apical dendrite has an explicit error representation by simultaneously integrating top-down excitation and lateral inhibition. As a consequence, we do not need to postulate separate temporal phases, and our network operates continuously while plasticity at all synapses is always turned on.

Error minimization is an integral part of brain function according to predictive coding theories (Rao and Ballard, 1999; Friston, 2005). Interestingly, recent work has shown that backprop can be mapped onto a predictive coding network architecture (Whittington and Bogacz, 2017), related to the general framework introduced by LeCun (1988). A possible network implementation is suggested by Whittington and Bogacz (2017) that requires intricate circuitry with appropriately tuned error-representing neurons. According to this work, the only plastic synapses are those that connect prediction and error neurons. By contrast, in our model, lateral, bottom-up and top-down connections are all plastic, and errors are directly encoded in dendritic compartments.

# 3 Results

## 3.1 Learning in dendritic error networks approximates backprop

In our model, neurons implicitly carry and transmit errors across the network. In the supplementary material, we formally show such propagation of errors for networks in a particular regime, which we term *self-predicting*. Self-predicting nets are such that when no external target is provided to output layer neurons, the lateral input from interneurons cancels the internally generated top-down feedback and renders apical dendrites silent. In this case, the output becomes a feedforward function of the input, which can in theory be optimized by conventional backprop. We demonstrate that synaptic plasticity in self-predicting nets approximates the weight changes prescribed by backprop.

We summarize below the main points of the full analysis (see SM). First, we show that somatic membrane potentials at hidden layer $k$ integrate feedforward predictions (encoded in basal dendritic potentials) with backpropagated errors (encoded in apical dendritic potentials):

$$\mathbf{u}_k^{\mathrm{P}} = \mathbf{u}_k^- + \lambda^{N-k+1} \mathbf{W}_{k,k+1}^{\mathrm{PP}} \left( \prod_{l=k+1}^{N-1} \mathbf{D}_l^- \mathbf{W}_{l,l+1}^{\mathrm{PP}} \right) \mathbf{D}_N^- \left( \mathbf{u}_N^{\mathrm{trgt}} - \mathbf{u}_N^- \right) + \mathcal{O}(\lambda^{N-k+2}).$$

Parameter $\lambda \ll 1$ sets the strength of feedback and teaching versus bottom-up inputs and is assumed to be small to simplify the analysis. The first term is the basal contribution and corresponds to $\mathbf{u}_k^-$, the activation computed by a purely feedforward network that is obtained by removing lateral and top-down weights from the model (here and below, we use superscript '-' to refer to the feedforward model). The second term (of order $\lambda^{N-k+1}$) is an error that is backpropagated from the output layer down to $k$-th layer hidden neurons; matrix $\mathbf{D}_k$ is a diagonal matrix with $i$-th entry containing the derivative of the neuronal transfer function evaluated at $u_{k,i}^-$.

Second, we compare model synaptic weight updates for the bottom-up connections to those prescribed by backprop. Output layer updates are exactly equal by construction. For hidden neuron synapses,

we obtain

$$\Delta\mathbf{W}_{k,k-1}^{\mathrm{PP}} = \eta_{k,k-1}^{\mathrm{PP}} \lambda^{N-k+1} \left( \prod_{l=k}^{N-1} \mathbf{D}_l^- \mathbf{W}_{l,l+1}^{\mathrm{PP}} \right) \mathbf{D}_N^- \left(\mathbf{u}_N^{\mathrm{trgt}} - \mathbf{u}_N^-\right) \left(\mathbf{r}_{k-1}^-\right)^T + \mathcal{O}(\lambda^{N-k+2}).$$

Up to a factor which can be absorbed in the learning rate, this plasticity rule becomes equal to the backprop weight change in the weak feedback limit $\lambda \to 0$, provided that the top-down weights are set to the transpose of the corresponding feedforward weights.

In our simulations, top-down weights are either set at random and kept fixed, in which case the equation above shows that the plasticity model optimizes the predictions according to an approximation of backprop known as feedback alignment (Lillicrap et al., 2016); or learned so as to minimize an inverse reconstruction loss, in which case the network implements a form of target propagation (Bengio, 2014; Lee et al., 2015).

### 3.2 Deviations from self-predictions encode backpropagated errors

To illustrate learning in the model and to confirm our analytical insights we first study a very simple task: memorizing a single input-output pattern association with only one hidden layer; the task naturally generalizes to multiple memories.

Given a self-predicting network (established by microcircuit plasticity, Fig. S1, see SM for more details), we focus on how prediction errors get propagated backwards when a novel teaching signal is provided to the output layer, modeled via the activation of additional somatic conductances in output pyramidal neurons. Here we consider a network model with an input, a hidden and an output layer (layers 0, 1 and 2, respectively; Fig. 1A).

When the pyramidal cell activity in the output layer is nudged towards some desired target (Fig. 1B (i)), the bottom-up synapses $\mathbf{W}_{2,1}^{\mathrm{PP}}$ from the lower layer neurons to the basal dendrites are adapted, again according to the plasticity rule that implements the dendritic prediction of somatic spiking (see Eq. 7). What these synapses cannot explain away encodes a dendritic error in the pyramidal neurons of the lower layer 1. In fact, the self-predicting microcircuit can only cancel the feedback that is produced by the lower layer activity.

The somatic integration of apical activity induces plasticity at the bottom-up synapses $\mathbf{W}_{1,0}^{\mathrm{PP}}$ (Eq. 7). As the apical error changes the somatic activity, plasticity of the $\mathbf{W}_{1,0}^{\mathrm{PP}}$ weights tries to further reduce the error in the output layer. Importantly, the plasticity rule depends only on local information available at the synaptic level: postsynaptic firing and dendritic branch voltage, as well as the presynaptic activity, in par with phenomenological models of synaptic plasticity (Sjöström et al., 2001; Clopath et al., 2010; Bono and Clopath, 2017). This learning occur concurrently with modifications of lateral interneuron weights which track changes in the output layer. Through the course of learning the network comes to a point where the novel top-down input is successfully predicted (Fig. 1B,C).

### 3.3 Network learns to solve a nonlinear regression task

We now test the learning capabilities of the model on a nonlinear regression task, where the goal is to associate sensory input with the output of a separate multilayer network that transforms the same sensory input (Fig. 2A). More precisely, a pyramidal neuron network of dimensions 30-50-10 (and 10 hidden layer interneurons) learns to approximate a random nonlinear function implemented by a held-aside feedforward network of dimensions 30-20-10. One teaching example consists of a randomly drawn input pattern $\mathbf{r}_0^{\mathrm{P}}$ assigned to corresponding target $\mathbf{r}_2^{\mathrm{trgt}} = \phi(k_{2,1}\mathbf{W}_{2,1}^{\mathrm{trgt}} \phi(k_{1,0} \mathbf{W}_{1,0}^{\mathrm{trgt}} \mathbf{r}_0^{\mathrm{P}}))$, with scale factors $k_{2,1} = 10$ and $k_{1,0} = 2$. Teacher network weights and input pattern entries are sampled from a uniform distribution $U(-1, 1)$. We used a soft rectifying nonlinearity as the neuronal transfer function, $\phi(u) = \gamma \log(1 + \exp(\beta(u - \theta)))$, with $\gamma = 0.1$, $\beta = 1$ and $\theta = 3$. This parameter setting led to neuronal activity in the nonlinear, sparse firing regime.

The network is initialized to a random initial synaptic weight configuration, with both pyramidal-pyramidal $\mathbf{W}_{1,0}^{\mathrm{PP}}$, $\mathbf{W}_{2,1}^{\mathrm{PP}}$, $\mathbf{W}_{1,2}^{\mathrm{PP}}$ and pyramidal-interneuron weights $\mathbf{W}_{1,1}^{\mathrm{IP}}$, $\mathbf{W}_{1,1}^{\mathrm{PI}}$ independently drawn from a uniform distribution. Top-down weight matrix $\mathbf{W}_{1,2}^{\mathrm{PP}}$ is kept fixed throughout, in the spirit of feedback alignment (Lillicrap et al., 2016). Output layer teaching currents $\mathbf{i}_2^{\mathrm{P}}$ are set so as to nudge $\mathbf{u}_2^{\mathrm{P}}$ towards the teacher-generated $\mathbf{u}_2^{\mathrm{trgt}}$. Learning rates were manually chosen to yield best

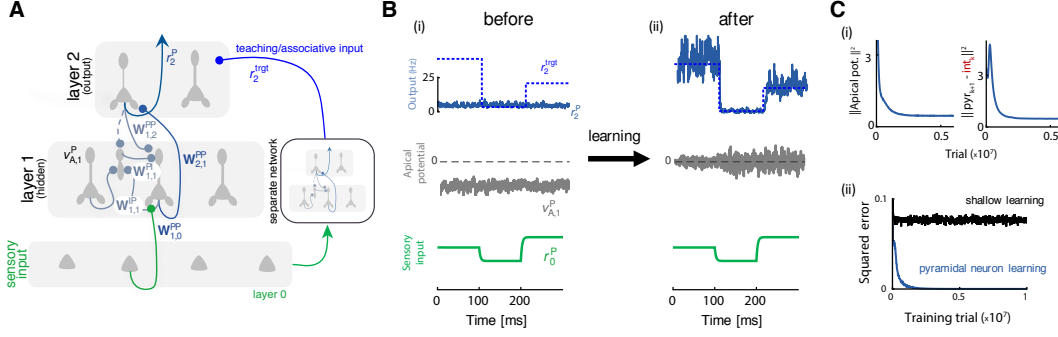

Figure 2: **Dendritic error microcircuit learns to solve a nonlinear regression task online and without phases.** (**A-C**) Starting from a random initial weight configuration, a 30-50-10 fully-connected network learns to approximate a nonlinear function ('separate network') from input-output pattern pairs. (**B**) Example firing rates for a randomly chosen output neuron ($r_2^P$, blue noisy trace) and its desired target imposed by the associative input ($r_2^{trgt}$, blue dashed line), together with the voltage in the apical compartment of a hidden neuron ($v_{A,1}^P$, grey noisy trace) and the input rate from the sensory neuron ($r_0^P$, green). Traces are shown before (i) and after learning (ii). (**C**) Error curves for the full model and a shallow model for comparison.

performance. Some learning rate tuning was required to ensure the microcircuit could track the changes in the bottom-up pyramidal-pyramidal weights, but we did not observe high sensitivity once the correct parameter regime was identified. Error curves are exponential moving averages of the sum of squared errors loss $\|\mathbf{r}_2^P - \mathbf{r}_2^{trgt}\|^2$ computed after every example on unseen input patterns. Test error performance is measured in a noise-free setting ($\sigma = 0$). Plasticity induction terms given by Eqs. 7-9 are low-pass filtered with time constant $\tau_w$ before being definitely consolidated, to dampen fluctuations; synaptic plasticity is kept on throughout. Plasticity and neuron model parameters are as defined above.

We let learning occur in continuous time without pauses or alternations in plasticity as input patterns are sequentially presented. This is in contrast to previous learning models that rely on computing activity differences over distinct phases, requiring temporally nonlocal computation, or globally coordinated plasticity rule switches (Hinton and McClelland, 1988; O'Reilly, 1996; Xie and Seung, 2003; Scellier and Bengio, 2017; Guerguiev et al., 2017). Furthermore, we relaxed the bottom-up vs. top-down weight symmetry imposed by backprop and kept the top-down weights $\mathbf{W}_{1,2}^{PP}$ fixed.

Forward $\mathbf{W}_{1,2}^{PP}$ weights quickly aligned to $\sim 45^o$ of the feedback weights $\left(\mathbf{W}_{2,1}^{PP}\right)^T$ (see Fig. S1), in line with the recently discovered feedback alignment phenomenon (Lillicrap et al., 2016). This simplifies the architecture, because top-down and interneuron-to-pyramidal synapses need not be changed. We set the scale of the top-down weights, apical and somatic conductances such that feedback and teaching inputs were strong, to test the model outside the weak feedback regime ($\lambda \to 0$) for which our SM theory was developed. Finally, to test robustness, we injected a weak noise current to every neuron.

Our network was able to learn this harder task (Fig. 2B), performing considerably better than a shallow learner where only hidden-to-output weights were adjusted (Fig. 2C). Useful changes were thus made to hidden layer bottom-up weights. The self-predicting network state emerged throughout learning from a random initial configuration (see SM; Fig. S1).

### 3.4 Microcircuit network learns to classify handwritten digits

Next, we turn to the problem of classifying MNIST handwritten digits. We wondered how our model would fare in this benchmark, in particular whether the prediction errors computed by the interneuron microcircuit would allow learning the weights of a hierarchical nonlinear network with multiple hidden layers. To that end, we trained a deeper, larger 4-layer network (with 784-500-500-10 pyramidal neurons, Fig. 3A) by pairing digit images with teaching inputs that nudged the 10 output neurons towards the correct class pattern. We initialized the network to a random but self-predicting

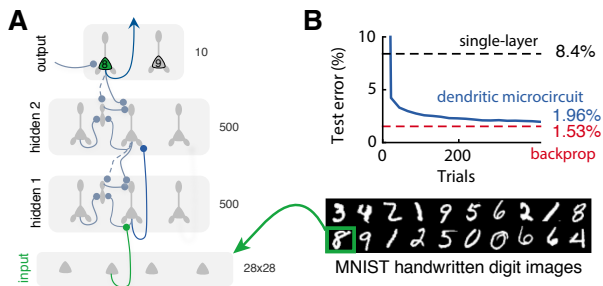

Figure 3: **Dendritic error networks learn to classify handwritten digits.** (**A**) A network with two hidden layers learns to classify handwritten digits from the MNIST data set. (**B**) Classification error achieved on the MNIST testing set (blue; cf. shallow learner (black) and standard backprop[6](red)).

configuration where interneurons cancelled top-down inputs, rendering the apical compartments silent before training started. Top-down and interneuron-to-pyramidal weights were kept fixed.

Here for computational efficiency we used a simplified network dynamics where the compartmental potentials are updated only in two steps before applying synaptic changes. In particular, for each presented MNIST image, both pyramidal and interneurons are first initialized to their bottom-up prediction state (3), $\mathbf{u}_k = \mathbf{v}_{\mathrm{B},k}$, starting from layer 1 up to the top layer $N$. Output layer neurons are then nudged towards their desired target $\mathbf{u}_N^{\mathrm{trgt}}$, yielding updated somatic potentials $\mathbf{u}_N^{\mathrm{P}} = (1 - \lambda_N)\,\mathbf{v}_{\mathrm{B},N} + \lambda_N\,\mathbf{u}_N^{\mathrm{trgt}}$. To obtain the remaining final compartmental potentials, the network is visited in reverse order, proceeding from layer $k = N - 1$ down to $k = 1$. For each $k$, interneurons are first updated to include top-down teaching signals, $\mathbf{u}_k^{\mathrm{I}} = (1 - \lambda_I)\,\mathbf{v}_k^{\mathrm{I}} + \lambda_I\,\mathbf{u}_{k+1}^{\mathrm{P}}$; this yields apical compartment potentials according to (4), after which we update hidden layer somatic potentials as a convex combination with mixing factor $\lambda_k$. The convex combination factors introduced above are directly related to neuron model parameters as conductance ratios. Synaptic weights are then updated according to Eqs. 7-10. Such simplified dynamics approximates the full recurrent network relaxation in the deterministic setting $\sigma \to 0$, with the approximation improving as the top-down dendritic coupling is decreased, $g_{\mathrm{A}} \to 0$.

We train the models on the standard MNIST handwritten image database, further splitting the training set into 55000 training and 5000 validation examples. The reported test error curves are computed on the 10000 held-aside test images. The four-layer network shown in Fig. 3 is initialized in a self-predicting state with appropriately scaled initial weight matrices. For our MNIST networks, we used relatively weak feedback weights, apical and somatic conductances (see SM) to justify our simplified approximate dynamics described above, although we found that performance did not appreciably degrade with larger values. To speed-up training we use a mini-batch strategy on every learning rule, whereby weight changes are averaged across 10 images before being applied. We take the neuronal transfer function $\phi$ to be a logistic function, $\phi(u) = 1/(1 + \exp(-u))$ and include a learnable threshold on each neuron, modelled as an additional input fixed at unity with a plastic weight. Desired target class vectors are 1-hot coded, with $r_N^{\mathrm{trgt}} \in \{0.1, 0.8\}$. During testing, the output is determined by picking the class label corresponding to the neuron with highest firing rate. We found the model to be relatively robust to learning rate tuning on the MNIST task, except for the rescaling by the inverse mixing factor to compensate for teaching signal dilution (see SM for the exact parameters).

The network was able to achieve a test error of 1.96%, Fig. 3B, a figure not overly far from the reference mark of non-convolutional artificial neural networks optimized with backprop (1.53%) and comparable to recently published results that lie within the range 1.6-2.4% (Lee et al., 2015; Lillicrap et al., 2016; Nøkland, 2016). The performance of our model also compares favorably to the 3.2% test error reported by Guerguiev et al. (2017) for a two-hidden-layer network. This was possible despite the asymmetry of forward and top-down weights and at odds with exact backprop, thanks to a feedback alignment dynamics. Apical compartment voltages remained approximately silent when output nudging was turned off (data not shown), reflecting the maintenance of a self-predicting state throughout learning, which enabled the propagation of errors through the network. To further demonstrate that the microcircuit was able to propagate errors to deeper hidden layers, and that the task was not being solved by making useful changes only to the weights onto the topmost hidden layer, we re-ran the experiment while keeping fixed the pyramidal-pyramidal weights connecting the two hidden layers. The network still learned the dataset and achieved a test error of 2.11%.

As top-down weights are likely plastic in cortex, we also trained a one-hidden-layer (784-1000-10) network where top-down weights were learned on a slow time-scale according to learning rule (10). This inverse learning scheme is closely related to target propagation (Bengio, 2014; Lee et al., 2015). Such learning could play a role in perceptual denoising, pattern completion and disambiguation, and boost alignment beyond that achieved by pure feedback alignment (Bengio, 2014). Starting from random initial conditions and keeping all weights plastic (bottom-up, lateral and top-down) throughout, our network achieved a test classification performance of 2.48% on MNIST. Once more, useful changes were made to hidden synapses, even though the microcircuit had to track changes in both the bottom-up and the top-down pathways.

## 4    Conclusions

Our work makes several predictions across different levels of investigation. Here we briefly highlight some of these predictions and related experimental observations. The most fundamental feature of the model is that distal dendrites encode error signals that instruct learning of lateral and bottom-up connections. While monitoring such dendritic signals during learning is challenging, recent experimental evidence suggests that prediction errors in mouse visual cortex arise from a failure to locally inhibit motor feedback (Zmarz and Keller, 2016; Attinger et al., 2017), consistent with our model. Interestingly, the plasticity rule for apical dendritic inhibition, which is central to error encoding in the model, received support from another recent experimental study (Chiu et al., 2018).

A further implication of our model is that prediction errors occurring at a higher-order cortical area would imply also prediction errors co-occurring at earlier areas. Recent experimental observations in the macaque face-processing hierarchy support this (Schwiedrzik and Freiwald, 2017).

Here we have focused on the role of a specific interneuron type (SST) as a feedback-specific interneuron. There are many more interneuron types that we do not consider in our framework. One such type are the PV (parvalbumin-positive) cells, which have been postulated to mediate a somatic excitation-inhibition balance (Vogels et al., 2011; Froemke, 2015) and competition (Masquelier and Thorpe, 2007; Nessler et al., 2013). These functions could in principle be combined with our framework in that PV interneurons may be involved in representing another type of prediction error (e.g., generative errors).

Humans have the ability to perform fast (e.g., one-shot) learning, whereas neural networks trained by backpropagation of error (or approximations thereof, like ours) require iterating over many training examples to learn. This is an important open problem that stands in the way of understanding the neuronal basis of intelligence. One possibility where our model naturally fits is to consider multiple subsystems (for example, the neocortex and the hippocampus) that transfer knowledge to each other and learn at different rates (McClelland et al., 1995; Kumaran et al., 2016).

Overall, our work provides a new view on how the brain may solve the credit assignment problem for time-continuous input streams by approximating the backpropagation algorithm, and bringing together many puzzling features of cortical microcircuits.

### Acknowledgements

The authors would like to thank Timothy P. Lillicrap, Blake Richards, Benjamin Scellier and Mihai A. Petrovici for helpful discussions. WS thanks Matthew Larkum for many inspiring discussions on dendritic processing. JS thanks Elena Kreutzer, Pascal Leimer and Martin T. Wiechert for valuable feedback and critical reading of the manuscript.

This work has been supported by the Swiss National Science Foundation (grant 310030L-156863 of WS), the European Union's Horizon 2020 Framework Programme for Research and Innovation under the Specific Grant Agreement No. 785907 (Human Brain Project), NSERC, CIFAR, and Canada Research Chairs.

## Footnotes

[4]Note that in biology a target may be represented by an associative signal from the motor cortex to a sensory cortex (Attinger et al., 2017).

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
