[Supplementary Material · neurips_dendritic_error_networks_camera_ready_supp.pdf]

# Supplementary Material: Dendritic cortical microcircuits approximate the backpropagation algorithm

**The dendritic cortical circuit learns to predict self-generated top-down input**

Figure S1: **Dendritic cortical circuit learns to predict self-generated top-down input.** (**A**) Illustration of multilayer network architecture. The network consists of an input layer 0 (e.g., thalamic input), one or more intermediate (hidden) layers (represented by layer $k$ and layer $k+1$, which can be mapped onto primary and higher sensory layers) and an output layer $N$ (e.g., motor cortex) (left). Each hidden layer consists of a microcircuit with pyramidal cells and lateral inhibitory interneurons (e.g., SST cells) (right). Pyramidal cells consist of three compartments: a basal compartment (with voltage $\mathbf{v}_{B,k}^P$) that receives bottom-up input; an apical compartment (with voltage $\mathbf{v}_{A,k}^P$), where top-down input converges to; and a somatic compartment (with voltage $\mathbf{u}_k^P$), that integrates the basal and apical voltage. Interneurons receive input from lateral pyramidal cells onto their own basal dendrites (with voltage $\mathbf{v}_{B,k}^I$), integrate this input on their soma (with voltage $\mathbf{u}_k^I$) and project back to the apical compartments (with voltage $\mathbf{v}_{A,k}^P$) of same-layer pyramidal cells. (**B**) In a pre-learning developmental stage, the network learns to predict and cancel top-down feedback given randomly generated inputs. Only pyramidal-to-interneuron synapses ($\mathbf{W}_{k,k}^{IP}$) and interneuron-to-pyramidal synapses ($\mathbf{W}_{k,k}^{PI}$) are changed at that stage according to predictive synaptic plasticity rules (defined in Eqs. 8 and 9). Example voltage traces for a randomly chosen downstream neuron ($\mathbf{u}_{k+1}^P$) and a corresponding interneuron ($\mathbf{u}_k^I$), a pyramidal cell apical compartment ($\mathbf{v}_{A,k}^P$) and an input neuron ($\mathbf{u}_0^P$), before (i) and after (ii) development, for three consecutively presented input patterns. Once learning of the lateral synapses from and onto interneurons has converged, self-generated top-down signals are predicted by the network — it is in a *self-predicting state*. Here we use a concrete network with one hidden layer and 30-20-10 pyramidal neurons (input-hidden-output). Note that no desired targets are presented to the output layer (cf. Fig. 1); the network is solely driven by random inputs. (**C**) Lateral inhibition cancels top-down input. (i) Interneurons learn to match next-layer pyramidal neuron activity as their input weights $\mathbf{W}_{k,k}^{IP}$ adapt (see main text for details). (ii) Concurrently, learning of interneuron-to-pyramidal synapses ($\mathbf{W}_{k,k}^{PI}$) silences the apical compartment of pyramidal neurons, but pyramidal neurons remain active (cf. B). This is a general effect, as the lateral microcircuit learns to predict and cancel the expected top-down input for every random pattern.

The microcircuit model introduced in the main text is key to encode and backpropagate errors across the network. Here, we illustrate how synaptic plasticity of lateral interneuron connections establishes a network regime, which we term *self-predicting*, whereby lateral input cancels the self-generated top-down feedback, effectively silencing apical dendrites. For this reason, SST cells are functionally inhibitory and are henceforth referred to as interneurons. Crucially, when the circuit is in this so-called self-predicting state, presenting a novel external signal at the output layer gives rise to top-down activity that cannot be explained away by the interneuron circuit. Below we show that these apical mismatches between top-down and lateral input constitute backpropagated, neuron-specific errors that drive plasticity on the forward weights to the hidden pyramidal neurons.

Learning to predict the feedback signals involves adapting both weights from and to the lateral interneuron circuit. Consider a network that is driven by a succession of sensory input patterns

(Fig. S1B, bottom row). Learning to cancel the feedback input is divided between both the weights from pyramidal cells to interneurons, $\mathbf{W}^{\mathrm{IP}}_{k,k}$, and from interneurons to pyramidal cells, $\mathbf{W}^{\mathrm{PI}}_{k,k}$.

First, due to the somatic teaching feedback, learning of the $\mathbf{W}^{\mathrm{IP}}_{k,k}$ weights leads interneurons to better reproduce the activity of the respective higher layer $k+1$ (Fig. S1B (i)). A failure to reproduce layer $k+1$ activity generates an internal prediction error at the dendrites of the interneurons, which triggers synaptic plasticity (as defined by Eq. 8) that corrects for the wrong dendritic prediction and eventually leads to a faithful tracing of the upper layer activity by the lower layer interneurons (Fig. S1B (ii)). The mathematical analysis (see section below, Eq. 37) shows that the plasticity rule (8) makes the inhibitory population implement the same function of the layer-$k$ pyramidal cell activity as done by the layer–$(k+1)$ pyramidal neurons. Thus, the interneurons will learn to mimic the layer–$(k+1)$ pyramidal neurons (Fig. S1Ci).

Second, as the interneurons mirror upper layer activity, inter-to-pyramidal neuron synapses within the same layer ($\mathbf{W}^{\mathrm{PI}}_{k,k}$, Eq. 9) successfully learn to cancel the top-down input to the apical dendrite (Fig. S1Cii), independently of the actual input stimulus that drives the network. By doing so, the inter-to-pyramidal neuron weights $\mathbf{W}^{\mathrm{PI}}_{k,k}$ learn to mirror the top-down weights onto the lower layer pyramidal neurons. The learning of the weights onto and from the interneurons works in parallel: as the interneurons begin to predict the activity of pyramidal cells in layer $k+1$, it becomes possible for the plasticity at interneuron-to-pyramidal synapses (Eq. 9) to find a synaptic weight configuration which precisely cancels the top-down feedback (see also Eq. 39 below). At this stage, every pattern of activity generated by the hidden layers of the network is explained by the lateral circuitry, Fig. S1C (ii). Importantly, once learning of the lateral interneurons has converged, the apical input cancellation occurs irrespective of the actual bottom-up sensory input. Therefore, interneuron synaptic plasticity leads the network to a *self-predicting state*.

Figure S2: **Emergence and maintenance of a self-predicting network state while learning a target function.** (**A**, **B**) Starting from random initial conditions (see Fig. 2), co-evolving bottom-up pyramidal-pyramidal and lateral microcircuit pyramidal-interneuron synaptic weights lead the network to a self-predicting state. To quantify the approximation error, we used the squared Frobenius matrix norm. Pyramidal-to-interneuron $\mathbf{W}^{\mathrm{IP}}_{1,1}$ and apical-targetting weights $\mathbf{W}^{\mathrm{PI}}_{1,1}$ approach ideal values (cf. supplementary mathematical analysis, and Fig. S1 and Fig. 1), allowing the backpropagation of output errors to layer 1 neurons. This state is maintained throughout, as bottom-up weights learn the target function (Fig. 2C). (**C**) Quickly after learning starts, bottom-up and top-down pyramidal-pyramidal weights align, a phenomenon known as feedback alignment (Lillicrap et al., 2016); by virtue of simultaneous pyramidal and interneuron synaptic plasticity the network effectively learns how to backpropagate errors.

We propose that the emergence of this state could occur during development, consistent with experimental findings (Dorrn et al., 2010; Froemke, 2015). Starting from a cross-layer self-predicting configuration helps speed-up learning of specific tasks, but is not essential. Indeed, we were able to train a nonlinear regression model (cf. Fig. 2) and an MNIST network starting from random conditions. Appropriate tuning of learning rates quickly led the network to a self-predicting state, which unlocked learning of the task, see Fig. S2.

**Supplementary data**

Below we detail the model parameters used to generate the figures presented in the paper.

**Fig. S1 details**. The parameters for the compartmental model neuron were: $g_\mathrm{A} = 0.8$, $g_\mathrm{B} = g_\mathrm{D} = 1.0$, $g_\mathrm{lk} = 0.1$. Interneuron somatic teaching conductances were balanced to yield overall nudging strength $g_\mathrm{som} = 0.8$. Initial weight matrix entries were independently drawn from a uniform distribution $U(-1, 1)$. We used a soft rectifying transfer function $\phi(u) = \log(1 + \exp(u))$. We chose background activity levels of $\sigma = 0.1$. The learning rates were set as $\eta^\mathrm{IP}_{1,1} = 0.0002375$ and $\eta^\mathrm{PI}_{1,1} = 0.0005$.

Input patterns were smoothly transitioned by low-pass filtering $\mathbf{u}^\mathrm{P}_0$ with time constant $\tau_0 = 3$. A transition between patterns was triggered every 100 ms. Weight changes were low pass filtered with time constant $\tau_w = 30$. The dynamical equations were solved using Euler's method with a time step of 0.1, which resulted in 1000 integration time steps per pattern.

**Fig. 1 details**. We used learning rates $\eta^\mathrm{PP}_{1,0} = \eta^\mathrm{IP}_{1,1} = 0.0011875$ and $\eta^\mathrm{PP}_{2,1} = 0.0005$. Remaining parameters as used for Fig. S1.

**Fig. 2 details**. Initial forward and pyramidal-interneuron weights were drawn independently from a uniform distribution $U(-0.1, 0.1)$. The network learned under a background noise level of $\sigma = 0.3$. The learning rates were $\eta^\mathrm{IP}_{1,1} = 0.0011875$, $\eta^\mathrm{PI}_{1,1} = 0.0059375$, $\eta^\mathrm{PP}_{1,0} = 0.0011875$, $\eta^\mathrm{PP}_{2,1} = 0.0005$. Weight matrix $\mathbf{W}^\mathrm{PP}_{1,2}$ was kept fixed, so the model relied on a feedback alignment mechanism to learn. Remaining parameters as used for Fig. S1.

**Fig. 3 details**. We chose mixing factors $\lambda_3 = \lambda_I = 0.1$ and $\lambda_1 = \lambda_2 = 0.3$. Forward learning rates were $\eta^\mathrm{PP}_{3,2} = 0.001/\lambda_3$, $\eta^\mathrm{PP}_{2,1} = \eta^\mathrm{PP}_{3,2}/\lambda_2$, $\eta^\mathrm{PP}_{1,0} = \eta^\mathrm{PP}_{2,1}/\lambda_1$. Lateral learning rates were $\eta^\mathrm{IP}_{2,2} = 2\eta^\mathrm{PP}_{3,2}$ and $\eta^\mathrm{IP}_{1,1} = 2\eta^\mathrm{PP}_{2,1}$. Initial forward weights were drawn at random from a uniform distribution $U(-0.1, 0.1)$, and the remaining weights from $U(-1, 1)$.

**Supplementary analysis**

In this supplementary note we present a set of mathematical results concerning the network and plasticity model described in the main text.

To proceed analytically we make a number of simplifying assumptions. Unless noted otherwise, we study the network in a deterministic setting and consider the limiting case where lateral microcircuit synaptic weights match the corresponding forward weights:

$$\mathbf{W}_{k,k}^{\mathrm{PI}} = -\mathbf{W}_{k,k+1}^{\mathrm{PP}} \equiv \mathbf{W}_{k,k}^{\mathrm{PI}*} \tag{11}$$

$$\mathbf{W}_{k,k}^{\mathrm{IP}} = \frac{g_{\mathrm{B}} + g_{\mathrm{lk}}}{g_{\mathrm{B}} + g_{\mathrm{A}} + g_{\mathrm{lk}}} \mathbf{W}_{k+1,k}^{\mathrm{PP}} \equiv \mathbf{W}_{k,k}^{\mathrm{IP}*}, \tag{12}$$

The particular choice of proportionality factors, which depend on the neuron model parameters, is motivated below. Under the above configuration, the network becomes self-predicting.

To formally relate the encoding and propagation of errors implemented by the inhibitory microcircuit to the backpropagation of errors algorithm from machine learning, we consider the limit where top-down input is weak compared to the bottom-up drive. This limiting case results in error signals that decrease exponentially with layer depth, but allows us to proceed analytically.

We further assume that the top-down weights converging to the apical compartments are equal to the corresponding forward weights, $\mathbf{W}_{k,k+1}^{\mathrm{PP}} = \left(\mathbf{W}_{k+1,k}^{\mathrm{PP}}\right)^T$. Such weight symmetry is not essential for successful learning in a broad range of problems, as demonstrated in the main simulations and as observed before (Lee et al., 2015; Lillicrap et al., 2016; Nøkland, 2016). It is, however, required to frame learning as a gradient descent procedure. Furthermore, in the analyses of the learning rules, we assume that synaptic changes take place at a fixed point of the neuronal dynamics; we therefore consider discrete-time versions of the plasticity rules. This approximates the continuous-time plasticity model as long as changes in the inputs are slow compared to the neuronal dynamics.

For convenience, we will occasionally drop neuron type indices and refer to bottom-up weights $\mathbf{W}_{k+1,k}$ and to top-down weights $\mathbf{W}_{k,k+1}$. Additionally, we assume without loss of generality that the dendritic coupling conductance for interneurons is equal to the basal dendritic coupling of pyramidal neurons, $g_{\mathrm{D}} = g_{\mathrm{B}}$. Finally, whenever it is useful to distinguish whether output layer nudging is turned off, we use superscript '−'.

**Interneuron activity in the self-predicting state.** Following Urbanczik and Senn (2014), we note that steady state interneuron somatic potentials can be expressed as a convex combination of basal dendritic and pyramidal neuron potentials that are provided via somatic teaching input:

$$\mathbf{u}_k^{\mathrm{I}} = \frac{g_{\mathrm{B}}}{g_{\mathrm{lk}} + g_{\mathrm{B}} + g_{\mathrm{som}}} \mathbf{v}_k^{\mathrm{I}} + \frac{g_{\mathrm{som}}}{g_{\mathrm{lk}} + g_{\mathrm{B}} + g_{\mathrm{som}}} \mathbf{u}_{k+1}^{\mathrm{P}} = (1-\lambda)\,\hat{\mathbf{v}}_k^{\mathrm{I}} + \lambda\,\mathbf{u}_{k+1}^{\mathrm{P}}, \tag{13}$$

with $g_{\mathrm{B}}$ and $g_{\mathrm{lk}}$ the effective dendritic transfer and leak conductances, respectively, and $g_{\mathrm{som}}$ the total excitatory and inhibitory teaching conductance. In the equation above, $\hat{\mathbf{v}}_k^{\mathrm{I}} = \frac{g_{\mathrm{B}}}{g_{\mathrm{lk}} + g_{\mathrm{B}}} \mathbf{v}_k^{\mathrm{I}}$ is the interneuron dendritic prediction (cf. Eq. 8), and $\lambda \equiv \frac{g_{\mathrm{som}}}{g_{\mathrm{lk}} + g_{\mathrm{B}} + g_{\mathrm{som}}} \in [0, 1[$ is a mixing factor which controls the nudging strength for the interneurons. In other words, the current prediction $\hat{\mathbf{v}}_k^{\mathrm{I}}$ and the teaching signal are averaged with coefficients determined by normalized conductances. We will later consider the weak nudging limit of $\lambda \to 0$.

The relation $\hat{\mathbf{v}}_k^{\mathrm{I}} = \hat{\mathbf{v}}_{\mathrm{B},k+1}^{\mathrm{P}}$ holds when pyramidal-to-interneuron synaptic weights are equal to pyramidal-pyramidal forward weights, up to a scale factor: $\mathbf{W}_{k,k}^{\mathrm{IP}} = \frac{g_{\mathrm{lk}} + g_{\mathrm{B}}}{g_{\mathrm{lk}} + g_{\mathrm{B}} + g_{\mathrm{A}}} \mathbf{W}_{k+1,k}^{\mathrm{PP}}$, which simplifies to $\mathbf{W}_{N-1,N-1}^{\mathrm{IP}} = \mathbf{W}_{N,N-1}^{\mathrm{PP}}$ for the last layer where $g_{\mathrm{A}} = 0$ (to reduce clutter, we use the slightly abusive notation whereby $g_{\mathrm{A}}$ should be understood to be zero when referring to output layer neurons). This is the reason for the particular choice of ideal pyramidal-to-interneuron weights presented in the preamble. The network is then internally consistent, in the sense that the interneurons predict the model's own predictions, held by pyramidal neurons.

**Bottom-up predictions in the absence of external nudging.** We first study the situation where the input pattern $\mathbf{r}_0$ is stationary and the output layer teaching input is disabled, $\mathbf{i}_N^{\mathrm{P}} = 0$. We show that the fixed point of the network dynamics is a state where somatic voltages are equal to basal voltages,

up to a dendritic attenuation factor. In other words, the network effectively behaves as if it were feedforward, in the sense that it computes the same function as the corresponding network with equal bottom-up but no top-down or lateral connections.

Specifically, in the absence of external nudging (indicated by the $-$ in the superscript), the somatic voltages of pyramidal and interneuron are given by the bottom-up dendritic predictions,

$$\mathbf{u}_k^{\mathrm{P},-} = \hat{\mathbf{v}}_{\mathrm{B},k}^{\mathrm{P},-} \equiv \frac{g_{\mathrm{B}}}{g_{\mathrm{lk}} + g_{\mathrm{B}} + g_{\mathrm{A}}} \, \mathbf{W}_{k,k-1}^{\mathrm{PP}} \, \phi(\hat{\mathbf{v}}_{\mathrm{B},k-1}^{\mathrm{P},-}) \tag{14}$$

$$\mathbf{u}_k^{\mathrm{I},-} = \hat{\mathbf{v}}_k^{\mathrm{I},-} \equiv \frac{g_{\mathrm{B}}}{g_{\mathrm{lk}} + g_{\mathrm{B}}} \, \mathbf{W}_{k,k}^{\mathrm{IP}} \, \phi(\hat{\mathbf{v}}_{\mathrm{B},k}^{\mathrm{P},-}). \tag{15}$$

To show that Eq. 14 describes the state of the network, we start at the output layer and set Eq. 1 to zero. Because nudging is turned off, we observe that $\mathbf{u}_N^{\mathrm{P}}$ is equal to $\hat{\mathbf{v}}_{\mathrm{B},N}^{\mathrm{P},-}$ if layer $N-1$ also satisfies $\mathbf{u}_{N-1}^{\mathrm{P}} = \hat{\mathbf{v}}_{\mathrm{B},N-1}^{\mathrm{P},-}$. The same recursively applies to the hidden layer below when its apical voltage vanishes, $\mathbf{v}_{\mathrm{A},N-1}^{\mathrm{P}} = 0$. Now we note that at the fixed point the interneuron cancels the corresponding pyramidal neuron, due to the assumption that the network is in a self-predicting state, which yields $\mathbf{u}_{N-1}^{\mathrm{I}} = \mathbf{u}_N^{\mathrm{P}}$. Together with the fact that $\mathbf{W}_{N-1,N-1}^{\mathrm{PI}} = -\mathbf{W}_{N-1,N}^{\mathrm{PP}}$, we conclude that the interneuron contribution to the apical compartment cancels the top-down pyramidal neuron input, yielding the required condition $\mathbf{v}_{\mathrm{A},N-1}^{\mathrm{P}} = 0$.

The above argument can be iterated down to the input layer, where activity is constant, and we arrive at Eq. 14.

**Zero plasticity induction in the absence of nudging.** In view of Eq. 14, which states that in the absence of external nudging the somatic voltages correspond to the basal predictions, no synaptic changes are induced in basal synapses on the pyramidal and interneurons as defined by the plasticity rules (7) and (8), respectively. Similarly, the apical voltages are equal to rest, $\mathbf{v}_{\mathrm{A},k}^{\mathrm{P},-} = \mathbf{v}_{\mathrm{rest}}$, when the top-down input is fully predicted, and no synaptic plasticity is induced in the inter-to-pyramidal neuron synapses, see (9). When noisy background currents are present, the average prediction error is zero, while momentary fluctuations will still trigger plasticity. Note that the above holds when the dynamics is away from equilibrium, under the additional constraint that the integration time constant of interneurons matches that of pyramidal neurons.

**Recursive prediction error propagation.** Prediction errors arise in the model whenever lateral interneurons cannot fully explain top-down input, leading to a deviation from baseline in apical dendrite activity. Here, we look at the network steady state equations for a stationary input pattern $\mathbf{r}_0$ and derive an iterative relationship which establishes the propagation across the network of prediction mismatches originating downstream. The following compartmental potentials are thus evaluated at a fixed point of the neuronal dynamics.

Under the assumption (11) of matching interneuron-to-pyramidal top-down weights, apical compartment potentials simplify to

$$\mathbf{v}_{\mathrm{A},k}^{\mathrm{P}} = \mathbf{W}_{k,k+1} \left[ \phi(\mathbf{u}_{k+1}^{\mathrm{P}}) - \phi(\mathbf{u}_k^{\mathrm{I}}) \right] = \mathbf{W}_{k,k+1} \, \mathbf{e}_{k+1}, \tag{16}$$

where we introduced error vector $\mathbf{e}_{k+1}$ defined as the difference between pyramidal and interneuron firing rates. Such deviation can be intuitively understood as an layer-wise interneuron prediction mismatch, being zero when interneurons perfectly explain pyramidal neuron activity. We now evaluate this difference vector at a fixed point to obtain a recurrence relation that links consecutive layers.

The steady-state somatic potentials of hidden pyramidal neurons are given by

$$\mathbf{u}_k^{\mathrm{P}} = \frac{g_{\mathrm{B}}}{g_{\mathrm{lk}} + g_{\mathrm{B}} + g_{\mathrm{A}}} \mathbf{v}_{\mathrm{B},k}^{\mathrm{P}} + \frac{g_{\mathrm{A}}}{g_{\mathrm{lk}} + g_{\mathrm{B}} + g_{\mathrm{A}}} \mathbf{v}_{\mathrm{A},k}^{\mathrm{P}} = \hat{\mathbf{v}}_{\mathrm{B},k}^{\mathrm{P}} + \lambda \, \mathbf{v}_{\mathrm{A},k}^{\mathrm{P}}$$
$$= \hat{\mathbf{v}}_{\mathrm{B},k}^{\mathrm{P}} + \lambda \, \mathbf{W}_{k,k+1} \, \mathbf{e}_{k+1}. \tag{17}$$

To shorten the following, we assumed that the apical attenuation factor is equal to the interneuron nudging strength $\lambda$. As previously mentioned, we proceed under the assumption of weak feedback, $\lambda$ small. As for the corresponding interneurons, we insert Eq. 17 into Eq. 13 and note that when the

network is in a self-predicting state we have $\hat{\mathbf{v}}_{k-1}^{\mathrm{I}} = \hat{\mathbf{v}}_{\mathrm{B},k}^{\mathrm{P}}$, yielding

$$\mathbf{u}_{k-1}^{\mathrm{I}} = (1-\lambda)\,\hat{\mathbf{v}}_{\mathrm{B},k}^{\mathrm{P}} + \lambda\left(\hat{\mathbf{v}}_{\mathrm{B},k}^{\mathrm{P}} + \lambda\,\mathbf{v}_{\mathrm{A},k}^{\mathrm{P}}\right) = \hat{\mathbf{v}}_{\mathrm{B},k}^{\mathrm{P}} + \lambda^2\,\mathbf{v}_{\mathrm{A},k}^{\mathrm{P}}. \tag{18}$$

Using the identities (17) and (18), we now expand to first order the difference vector $\mathbf{e}_k$ around $\hat{\mathbf{v}}_{\mathrm{B},k}^{\mathrm{P}}$ as follows

$$\mathbf{e}_k = \phi(\mathbf{u}_k^{\mathrm{P}}) - \phi(\mathbf{u}_{k-1}^{\mathrm{I}}) = \lambda\,\mathbf{D}_k\,\mathbf{v}_{\mathrm{A},k}^{\mathrm{P}} + \mathcal{O}\!\left(\lambda^2\,\|\mathbf{v}_{\mathrm{A},k}^{\mathrm{P}}\|\right). \tag{19}$$

Matrix $\mathbf{D}_k$ is a diagonal matrix with diagonal equal to $\phi'(\hat{\mathbf{v}}_{\mathrm{B},k}^{\mathrm{P}})$, i.e., whose $i$-th element reads $\frac{d\phi}{dv}(\hat{v}_{\mathrm{B},k,i}^{\mathrm{P}})$. It contains the derivative of the neuronal transfer function $\phi$ evaluated component-wise at the bottom-up predictions $\hat{\mathbf{v}}_{\mathrm{B},k+1}^{\mathrm{P}}$. Recalling Eq. 16, we obtain a recurrence relation

$$\mathbf{e}_k = \lambda\,\mathbf{D}_k\,\mathbf{W}_{k,k+1}\,\mathbf{e}_{k+1} + \mathcal{O}\!\left(\lambda^2\,\|\mathbf{W}_{k,k+1}\,\mathbf{e}_{k+1}\|\right). \tag{20}$$

Finally, last layer pyramidal neurons provide the initial condition by being directly nudged towards the desired target $\mathbf{u}_N^{\mathrm{trgt}}$. Their membrane potentials can be written as

$$\mathbf{u}_N^{\mathrm{P}} = (1-\lambda)\,\hat{\mathbf{v}}_{\mathrm{B},N}^{\mathrm{P}} + \lambda\,\mathbf{u}_N^{\mathrm{trgt}}, \tag{21}$$

and this gives an estimate for the error in the output layer of the form

$$\mathbf{e}_N = \lambda\,\mathbf{D}_N\left(\mathbf{u}_N^{\mathrm{trgt}} - \hat{\mathbf{v}}_{\mathrm{B},N}^{\mathrm{P}}\right) + \mathcal{O}\!\left(\lambda^2\,\|\mathbf{u}_N^{\mathrm{trgt}} - \hat{\mathbf{v}}_{\mathrm{B},N}^{\mathrm{P}}\|\right), \tag{22}$$

where for simplicity we took the same mixing factor $\lambda$ for pyramidal output and interneurons. Then, for an arbitrary layer, assuming that the synaptic weights and the remaining fixed parameters do not scale with $\lambda$, we arrive at

$$\mathbf{e}_k = \lambda^{N-k+1}\left(\prod_{l=k}^{N-1}\mathbf{D}_l\,\mathbf{W}_{l,l+1}\right)\mathbf{D}_N\left(\mathbf{u}_N^{\mathrm{trgt}} - \hat{\mathbf{v}}_{\mathrm{B},N}^{\mathrm{P}}\right) + \mathcal{O}(\lambda^{N-k+2}). \tag{23}$$

Thus, steady state potentials of apical dendrites (cf. Eq. 16) recursively encode neuron-specific prediction errors that can be traced back to a mismatch at the output layer.

**Learning as approximate error backpropagation.** In the previous section we found that neurons implicitly carry and transmit error information across the network. We now show how the proposed synaptic plasticity model, when applied at a steady state of the neuronal dynamics, can be recast as an approximate gradient descent learning procedure.

More specifically, we compare our model against learning through backprop (Rumelhart et al., 1986) or approximations thereof (Lee et al., 2015; Lillicrap et al., 2016) the weights of the feedfoward multilayer network obtained by removing interneurons and top-down connections from the intact network. For this reference model, the activations $\mathbf{u}_k^-$ are by construction equal to the bottom-up predictions obtained in the full model when output nudging is turned off, $\mathbf{u}_k^- \equiv \hat{\mathbf{v}}_{\mathrm{B},k}^{\mathrm{P},-}$, cf. Eq. 14. Thus, optimizing the weights in the feedforward model is equivalent to optimizing the predictions of the full model.

We now assume that $\phi$ is monotonically increasing and define the loss function

$$\mathcal{L}\!\left(\mathbf{u}_N^-, \mathbf{u}_N^{\mathrm{trgt}}\right) = -\sum_{i=1}^{N_N}\int_0^{u_{N,i}^-}\phi\!\left((1-\lambda)\,\nu + \lambda\,u_{N,i}^{\mathrm{trgt}}\right) - \phi(\nu)\,d\nu, \tag{24}$$

where $N_N$ denotes the number of output neurons. $\mathcal{L}$ can be thought of as the multilayer, multi-output unit analogue of the loss function optimized by the single neuron model (Urbanczik and Senn, 2014), where it stems directly from the particular chosen form of the learning rule (7). The nudging strength parameter $\lambda \in [0, 1[$ allows controlling the mixing with the target and can be understood as an additional learning rate parameter. Albeit unusual in form, function $\mathcal{L}$ imposes a cost similar to an ordinary squared error loss. Importantly, it has a minimum when $\mathbf{u}_N^- = \mathbf{u}_N^{\mathrm{trgt}}$ and it is lower bounded. Furthermore, it is differentiable with respect to compartmental voltages (and synaptic weights). It is therefore suitable for gradient descent optimization. As a side remark, $\mathcal{L}$ integrates to a quadratic function when $\phi$ is linear.

Gradient descent proceeds by changing synaptic weights according to

$$\Delta \mathbf{W}_{k,k-1} = -\eta \, \frac{\partial \mathcal{L}}{\partial \mathbf{W}_{k,k-1}}. \tag{25}$$

The required partial derivatives can be efficiently computed by the backpropagation of errors algorithm. For the network architecture we study, this yields a learning rule of the form

$$\Delta \mathbf{W}_{k,k-1}^{\text{bp}} = \eta \, \mathbf{e}_k^- \, \phi(\mathbf{u}_{k-1}^-)^T. \tag{26}$$

The error factor $\mathbf{e}_k^-$ can be expressed recursively as follows:

$$\mathbf{e}_k^- = \begin{cases} \phi\big((1-\lambda)\,\mathbf{u}_N^- + \lambda\,\mathbf{u}_N^{\text{trgt}}\big) - \phi\big(\mathbf{u}_N^-\big) & \text{if } k = N, \\ \mathbf{D}_k^- \, \mathbf{W}_{k+1,k}^T \, \mathbf{e}_{k+1}^- & \text{otherwise,} \end{cases} \tag{27}$$

ignoring constant factors that depend on conductance ratios, which can be dealt with by redefining learning rates or backward pass weights. As in the previous section, matrix $\mathbf{D}_k^-$ is a diagonal matrix, with diagonal equal to $\phi'(\mathbf{u}_k^-)$.

We first compare the fixed point equations of the original network to the feedforward activations of the reference model. Starting from the bottom most hidden layer, using Eqs. 16, 17 and 23, we notice that $\mathbf{u}_1^P = \mathbf{u}_1^- + \lambda\,\mathbf{v}_{A,1}^P = \mathbf{u}_1^- + \mathcal{O}(\lambda^N)$, as the bottom-up input is the same in both cases. Inserting this into second hidden layer steady state potentials and linearizing the neuronal transfer function gives $\mathbf{u}_2^P = \mathbf{u}_2^- + \lambda\,\mathbf{v}_{A,2}^P + \mathcal{O}(\lambda^N) = \mathbf{u}_2^- + \mathcal{O}(\lambda^{N-1})$. This can be repeated and for an arbitrary layer and neuron type we find

$$\mathbf{u}_k^P = \mathbf{u}_k^- + \lambda\,\mathbf{v}_{A,k}^P + \mathcal{O}(\lambda^{N-k+2}) = \mathbf{u}_k^- + \mathcal{O}(\lambda^{N-k+1}) \tag{28}$$

$$\mathbf{u}_{k-1}^I = \mathbf{u}_k^- + \mathcal{O}(\lambda^{N-k+2}). \tag{29}$$

Writing Eq. 28 in the first form emphasizes that the apical contributions dominate $\mathcal{O}(\lambda\,\mathbf{v}_{A,k}^P) = \mathcal{O}(\lambda^{N-k+1})$ the bottom-up corrections, which are of order $\mathcal{O}(\lambda^{N-k+2})$.

Next, we prove that up to a factor and to first order the apical term in Eq. 28 represents the backpropagated error in the feedforward network, $\mathbf{e}_k^-$. Starting from the topmost hidden layer apical potentials, we reevaluate difference vector (22) using (28). Linearization of the neuronal transfer function gives

$$\mathbf{v}_{A,N-1}^P = \lambda\,\mathbf{W}_{N-1,N}\,\mathbf{D}_N^-\left(\mathbf{u}_N^{\text{trgt}} - \mathbf{u}_N^-\right) + \mathcal{O}(\lambda^2). \tag{30}$$

Inserting the expression above into Eq. 28 and using Eq. 29 the apical compartment potentials at layer $N-1$ can then be recomputed. This procedure can be iterated until the input layer is reached. In general form, somatic membrane potentials at hidden layer $k$ can be expressed as

$$\mathbf{u}_k^P = \mathbf{u}_k^- + \lambda\,\mathbf{v}_{A,k}^P + \mathcal{O}(\lambda^{N-k+2}) \tag{31}$$

$$= \mathbf{u}_k^- + \lambda^{N-k+1}\,\mathbf{W}_{k,k+1}\left(\prod_{l=k+1}^{N-1} \mathbf{D}_l^-\,\mathbf{W}_{l,l+1}\right)\mathbf{D}_N^-\left(\mathbf{u}_N^{\text{trgt}} - \mathbf{u}_N^-\right) + \mathcal{O}(\lambda^{N-k+2}). \tag{32}$$

This equation shows that, to leading order of $\lambda$, hidden neurons mix and propagate forward purely bottom-up predictions with top-down errors that are computed at the output layer and spread backwards.

We are now in position to compare model synaptic weight updates to the ones prescribed by backprop. Output layer updates are exactly equal by construction, $\Delta \mathbf{W}_{N,N-1} = \Delta \mathbf{W}_{N,N-1}^{\text{bp}}$. For pyramidal-to-pyramidal neuron synapses from hidden layer $k-1$ to layer $k$, we obtain

$$\begin{aligned}
\Delta \mathbf{W}_{k,k-1} &= \eta_{k,k-1}\left[\phi(\mathbf{u}_k^P) - \phi(\hat{\mathbf{v}}_{B,k}^P)\right]\left(\mathbf{r}_{k-1}^P\right)^T \\
&= \eta_{k,k-1}\left[\phi\left(\mathbf{u}_k^- + \lambda\,\mathbf{v}_{A,k}^P + \mathcal{O}(\lambda^{N-k+2})\right) - \phi(\mathbf{u}_k^-)\right]\left(\mathbf{r}_{k-1}^- + \mathcal{O}(\lambda^{N-k+2})\right)^T \\
&= \eta_{k,k-1}\lambda^{N-k+1}\left(\prod_{l=k}^{N-1} \mathbf{D}_l^-\,\mathbf{W}_{l,l+1}\right)\mathbf{D}_N^-\left(\mathbf{u}_N^{\text{trgt}} - \mathbf{u}_N^-\right)\left(\mathbf{r}_{k-1}^-\right)^T + \mathcal{O}(\lambda^{N-k+2}),
\end{aligned} \tag{33}$$

while backprop learning rule (26) can be written as

$$\Delta \mathbf{W}^{\mathrm{bp}}_{k,k-1} = \eta \, \lambda \, \left( \prod_{l=k}^{N-1} \mathbf{D}^-_l \, \mathbf{W}_{l,l+1} \right) \mathbf{D}^-_N \left( \mathbf{u}^{\mathrm{trgt}}_N - \mathbf{u}^-_N \right) \left( \mathbf{r}^-_{k-1} \right)^T + \mathcal{O}(\lambda^2), \qquad (34)$$

where we used that, to first order, the output layer error factor is $\mathbf{e}^-_N = \lambda \, \mathbf{D}^-_N \left( \mathbf{u}^{\mathrm{trgt}}_N - \mathbf{u}^-_N \right) + \mathcal{O}(\lambda^2)$. Hence, up to a factor of $\lambda^{N-k}$ which can be absorbed in the learning rate $\eta_{k,k-1}$, changes induced by synaptic plasticity are equal to the backprop learning rule (26) in the limit $\lambda \to 0$, provided that the top-down weights are set to the transpose of the corresponding feedforward weights, $\mathbf{W}_{k,k+1} = \mathbf{W}^T_{k+1,k}$. The 'quasi-feedforward' condition $\lambda \to 0$ has also been invoked to relate backprop to two-phase contrastive Hebbian learning in Hopfield networks (Xie and Seung, 2003).

**Interneuron plasticity.** The analyses of the previous sections relied on the assumption that the synaptic weights to and from interneurons were set to their ideal values, cf. Eqs. 11 and 12. We now study the plasticity of the lateral microcircuit synapses and show that, under mild conditions, learning rules (8) and (9) yield the desired synaptic weight matrices.

We first study the learning of pyramidal-to-interneuron synapses $\mathbf{W}^{\mathrm{IP}}_{k,k}$. To quantify the degree to which the weights deviate from their optimal setting, we introduce the convex loss function

$$\mathcal{L}^{\mathrm{IP}}_k = \frac{1}{2} \operatorname{Tr} \left\{ (\mathbf{W}^{\mathrm{IP}*}_{k,k} - \mathbf{W}^{\mathrm{IP}}_{k,k})^T (\mathbf{W}^{\mathrm{IP}*}_{k,k} - \mathbf{W}^{\mathrm{IP}}_{k,k}) \right\}, \qquad (35)$$

where $\operatorname{Tr}(\mathbf{M})$ denotes the trace of matrix $\mathbf{M}$ and $\mathbf{W}^{\mathrm{IP}*}_{k,k} = \frac{g_{\mathrm{B}} + g_{\mathrm{lk}}}{g_{\mathrm{B}} + g_{\mathrm{A}} + g_{\mathrm{lk}}} \mathbf{W}^{\mathrm{PP}}_{k+1,k}$, as defined in Eq. 12.

Starting from the pyramidal-to-interneuron synaptic plasticity rule (8), we express the interneuron somatic potential in convex combination form (13) and then expand to first order around $\hat{\mathbf{v}}^{\mathrm{I}}_k$,

$$\begin{aligned}
\Delta \mathbf{W}^{\mathrm{IP}}_{k,k} &= \eta^{\mathrm{IP}}_{k,k} \left( \phi(\mathbf{u}^{\mathrm{I}}_k) - \phi(\hat{\mathbf{v}}^{\mathrm{I}}_k) \right) (\mathbf{r}^{\mathrm{P}}_k)^T \\
&= \eta^{\mathrm{IP}}_{k,k} \lambda \, \mathbf{D}^{\mathrm{IP}}_k \left( \mathbf{u}^{\mathrm{P}}_{k+1} - \hat{\mathbf{v}}^{\mathrm{I}}_k \right) (\mathbf{r}^{\mathrm{P}}_k)^T + \mathcal{O}(\lambda^2) \\
&= \eta^{\mathrm{IP}}_{k,k} \lambda \, \frac{g_{\mathrm{B}}}{g_{\mathrm{lk}} + g_{\mathrm{B}}} \, \mathbf{D}^{\mathrm{IP}}_k \left( \mathbf{W}^{\mathrm{IP}*}_{k,k} - \mathbf{W}^{\mathrm{IP}}_{k,k} \right) \mathbf{Q}_k + \mathcal{O}(\lambda^2). \qquad (36)
\end{aligned}$$

Matrix $\mathbf{Q}_k = \mathbf{r}^{\mathrm{P}}_k (\mathbf{r}^{\mathrm{P}}_k)^T$ denotes the outer product, and $\mathbf{D}^{\mathrm{IP}}_k$ is a diagonal matrix with $i$-th diagonal entry equal to $\phi'(\hat{\mathbf{v}}^{\mathrm{I}}_{k,i})$.

For simplicity, we ignore fluctuations arising from the stochastic sequential presentation of patterns (Bottou, 1998) and look only at the expected synaptic dynamics[7]. We absorb irrelevant scale factors and to avoid a vanishing update we rescale the learning rate $\hat{\eta}^{\mathrm{IP}}_{k,k}$ by $\lambda^{-1}$. Then, taking the limit $\lambda \to 0$ as in the previous sections yields

$$\begin{aligned}
\mathrm{E}\left[ \Delta \mathbf{W}^{\mathrm{IP}}_{k,k} \right] &= \hat{\eta}^{\mathrm{IP}}_{k,k} \left( \mathbf{W}^{\mathrm{IP}*}_{k,k} - \mathbf{W}^{\mathrm{IP}}_{k,k} \right) \mathrm{E}\left[ \mathbf{D}^{\mathrm{IP}}_k \mathbf{Q}_k \right] \\
&= -\hat{\eta}^{\mathrm{IP}}_{k,k} \frac{\partial \mathcal{L}^{\mathrm{IP}}_k}{\partial \mathbf{W}^{\mathrm{IP}}_{k,k}} \, \mathrm{E}\left[ \mathbf{D}^{\mathrm{IP}}_k \mathbf{Q}_k \right]. \qquad (37)
\end{aligned}$$

The expectation is taken over the pattern ensemble. In the last equality above, we used the fact that the gradient of $\mathcal{L}^{\mathrm{IP}}_k$ with respect to the lateral weights $\mathbf{W}^{\mathrm{IP}}_{k,k}$ is given by the difference $\mathbf{W}^{\mathrm{IP}*}_{k,k} - \mathbf{W}^{\mathrm{IP}}_{k,k}$.

As long as the expectation on the right-hand side of (37) is positive definite, the synaptic dynamics is within 90º of the gradient and thus leads to the unique minimum of $\mathcal{L}^{\mathrm{IP}}_k$. This condition is easily met in practice. For linear neurons, it amounts to requiring that the correlation matrix $\mathrm{E}[\mathbf{Q}]$ is positive definite. In other words, the patterns have to span $\mathbb{R}^{N_k}$, with $N_k$ being the number of pyramidal neurons at layer $k$. This is fulfilled when uncorrelated background noise currents are present, and it is likely the case for deterministic networks solving nontrivial tasks. For nonlinear neurons with a saturating transfer function $\phi$, saturation can lead to a matrix numerically close to singular and slow down learning. A weight matrix initialization that sets the neurons operating far from saturation is therefore an appropriate choice.

A mathematical analysis of the coupled system defined by the various plasticity rules acting in concert is rather involved and beyond the scope of this note. However, the learning of apical-targetting interneuron-to-pyramidal synapses can be studied in isolation by invoking a separation of timescales argument. To proceed, we assume that pyramidal-to-interneuron synapses are ideally set, $\mathbf{W}_{k,k}^{\mathrm{IP}} = \mathbf{W}_{k,k}^{\mathrm{IP}*}$. In practice, this translates to a choice of a small effective learning rate for apical-targetting weights $\eta_{k,k}^{\mathrm{PI}}$. Note that, indirectly, this requirement also imposes a constraint on how fast top-down pyramidal-to-pyramidal weights $\mathbf{W}_{k,k+1}^{\mathrm{PP}}$ can evolve. This is the parameter regime explored in the main text simulations with plastic top-down weights.

We can then proceed in a similar manner to the previous analysis, as we briefly outline below. Recalling from Eq. 11 that $\mathbf{W}_{k,k}^{\mathrm{PI}*} = -\mathbf{W}_{k,k+1}^{\mathrm{PP}}$, we define the loss function

$$\mathcal{L}_k^{\mathrm{PI}} = \frac{1}{2} \operatorname{Tr} \left\{ (\mathbf{W}_{k,k}^{\mathrm{PI}*} - \mathbf{W}_{k,k}^{\mathrm{PI}})^T (\mathbf{W}_{k,k}^{\mathrm{PI}*} - \mathbf{W}_{k,k}^{\mathrm{PI}}) \right\}. \tag{38}$$

After some manipulation, as $\lambda \to 0$ the expected synaptic change can be written as

$$\mathrm{E}\left[\Delta \mathbf{W}_{k,k}^{\mathrm{PI}}\right] = -\eta_{k,k}^{\mathrm{PI}} \frac{\partial \mathcal{L}_k^{\mathrm{PI}}}{\partial \mathbf{W}_{k,k}^{\mathrm{PI}}} \, \mathrm{E}\left[\mathbf{r}_{k+1}^{\mathrm{P}} (\mathbf{r}_{k+1}^{\mathrm{P}})^T\right], \tag{39}$$

which leads us to conclude that the weights converge to the appropriate values, provided that the correlation matrix of layer $k+1$ activity patterns is positive definite.

## Footnotes

[7]This can be understood as a batch learning protocol, where weight changes are accumulated in the limit of many patterns before being effectively consolidated as a synaptic update.