[Reviews · NeurIPS 2018]

Reviewer 1



Update after rebuttal: This is a strong paper. The rebuttal has addressed several of my concerns. ____________________ Summary: This paper proposes a new biological implementation of an approximate backpropagation algorithm that relies on multi compartment neuron models. Using two compartments allows errors and activities to be represented within the same neuron. The overall procedure is similar to contrastive Hebbian learning and relies on weak top down feedback from an initial ‘self-predicting’ settled state, but unlike contrastive Hebbian learning does not require separate phases. Experimental results show that the method can attain reasonable results on MNIST. Major comments: This paper presents an interesting approach to approximately implementing backpropagation that relies on a mixture of dendritic compartments and specific circuitry motifs. This is a fundamentally important topic and the results would likely be of interest to many, even if the specific hypothesis turns out to be incorrect. The paper is admirably clear and easy to follow. The method would appear to suffer from a similar difficulty as contrastive Hebbian learning: it only faithfully reflects backpropagation when top down feedback is weak, but this means that weight updates become very small in a deep network. Correcting this by scaling up the weight updates by the inverse of the top down feedback strength is problematic because this would require high precision and would drastically amplify small noise in the neural activity. Most of the experiments presented in the paper focus on shallow, single hidden layer networks. The MNIST experiment with a deeper two hidden layer network might provide an opportunity to test the robustness of the system by showing training curves with frozen first layer weights. More broadly, recent results have shown that schemes like feedback alignment often work well on simple datasets in relatively shallow networks, but struggle to match the performance of backprop in deeper networks on more complex datasets. The paper could be greatly strengthened by testing whether the proposed method really solves the credit assignment problem in substantially deep networks. In general, the test error rates reported here are typically in the range of what is achievable by a SVM which does not learn its internal representation (e.g., SVM with Gaussian kernel gets ~1.4%). Because of this it is important to show that the learning performance is critically dependent on learning the internal representation. It would be useful to report learning curves with random but frozen first layers (with optimized variance), in addition to the shallow network performance.

Reviewer 2



The authors develop a biologically plausible network model with synaptic plasticity that is able to learn specified tasks in a manner that approximates the backpropagation algorithm. The authors provide mathematical derivations showing the similarity between their biologically-plausible model and backpropagation, and convincing results for the performance of their model under a couple of tasks (nonlinear regression task, and classification of handwritten digits). This work is technically sound and will be a valuable contribution to the conference, in particular for the neuroscience inclined. A few comments: -the manuscript provides a comparison of the proposed model with a machine learning model (backpropagation in a densely connected network) in the MNIST classification task. For completeness, it would be important to also show the performance of the Guerguiev et al. 2017 model (that the authors cite), another biologically plausible model attempting to implement backpropagation; -the authors provide evidence for the biological plausibility of the model, by citing key experimental findings in neuroscience in recent years (see section Conclusions). In particular, the authors discuss recent work on connectivity and electrophysiology that is consistent with the implementation details of their model (connectivity between areas, cell types, neural activity signaling prediction error...). It would be interesting if the authors could also comment on how much data and how many trials this model needs to achieve good performance, and, if possible, how biologically plausible that is; -the authors should comment on the choice of hyperparameters across tasks (learning rates, noise magnitudes, numbers of neurons in the hidden layers): why the particular choices, and whether different parametrisations affect the performance. -- after rebuttal -- The authors' feedback addressed satisfactorily all my comments, and I therefore maintain my support for the acceptance of this paper.

Reviewer 3



The paper investigated how backpropagation can be implemented in neural systems. The authors studied a local hierarchical circuit with detailed dendritic structure. They showed that the dendrites compute the prediction error to effectively realize BP. This is a very interesting work, agreeing with the predictive coding idea and the related experiment evidence. It proposes a strategy to implement BP much more biologically plausible than others in the literature. The proposed strategy contains two interesting points: 1) the neural circuit does not require separate phases for activity propagation, error encoding, and error propagation, all done at the same time; 2) synaptic learning is driven by local dendritic prediction errors. Some comments: (1) In the analysis, according to equation (13), interneurons in every layer will get top-down somatic teacher signals from the corresponding pyramidal neurons, which is necessary to make the recursive formation of the predictive error signals propagation. However, in the regression and classification tasks, it seems only the output pyramidal neurons receive somatic teacher input and interneurons do not receive such a top-down nudging signal according to the parameter setting. Right?Dose this make a difference for the final performance with and without the top-down nudging signals to interneurons? (2) In the regression task, the top-down pyramidal-to-pyramidal weights are fixed; in the classification task, the top-down and interneuron-to-pyramidal weights were fixed. Are these setting crucial to successful training? If so, does this mean that our brain needs some control mechanisms to close or open the synaptic plasticity windows and the learning order of different connection weights types, such as learning the weights between interneurons and pyramidal neurons first and the bottom-up weights followed? I become more confident with the importance of this study, and decide to raise the score to 9.